# mRNA vaccines encoding fusion proteins of monkeypox virus antigens protect mice from *vaccinia virus* challenge

Fujun Hou[1,2,6], Yuntao Zhang[3,4,6], Xiaohu Liu [5,6], Yanal M Murad[5], Jiang Xu[1], Zhibin Yu[1], Xianwu Hua[1], Yingying Song[1], Jun Ding[1], Hongwei Huang[1,2,3,5], Ronghua Zhao[1,3,5], William Jia [1,3,5] ✉ & Xiaoming Yang [4] ✉

The recent outbreaks of mpox have raised concerns over the need for effective vaccines. However, the current approved vaccines have either been associated with safety concerns or are in limited supply. mRNA vaccines, which have shown high efficacy and safety against *SARS-CoV-2* infection, are a promising alternative. In this study, three mRNA vaccines are developed that encode *monkeypox virus* (*MPXV*) proteins A35R and M1R, including A35R extracellular domain -M1R fusions (VGPox 1 and VGPox 2) and a mixture of encapsulated full-length mRNAs for A35R and M1R (VGPox 3). All three vaccines induce early anti-A35R antibodies in female *Balb/c* mice, but only VGPox 1 and 2 generate detectable levels of anti-M1R antibodies at day 7 after vaccination. However, all three mRNA vaccine groups completely protect mice from a lethal dose of *vaccinia virus* (*VACV*) challenge. A single dose of VGPox 1, 2, and 3 provide protection against the lethal viral challenge within 7 days post-vaccination. Long-term immunity and protection were also observed in all three candidates. Additionally, VGPox 2 provided better passive protection. These results suggest that the VGPox series vaccines enhance immunogenicity and can be a viable alternative to current whole-virus vaccines to defend against mpox.

*Monkeypox virus* (*MPXV*) belongs to the Orthopoxvirus (OPXV) genus of the Poxviridae family, which also includes *variola virus* (*smallpox*) and *vaccinia virus* (*VACV*)[1]. Outbreaks of the *variola* virus caused millions of deaths until its global eradication in the 1980s, thanks to worldwide vaccination with live-virus preparations of the infectious *VACV*s.

The World Health Organization (WHO) declared the mpox outbreak a global health emergency between July 23, 2022, and May 11, 2023, due to a significant increase in the number of mpox infections worldwide[2]. The end of routine smallpox vaccinations may be one of the reasons for the recent outbreak[3], as *MPXV* and *smallpox* share highly homologous genomes, and antibodies against smallpox have shown significant cross-protection against *MPXV*[4–6]. Currently, there are two pox vaccines available: ACAM2000, an attenuated live VACV vaccine[7]; and JYNNEOS, a live but non-replicating virus[8]. The JYNNEOS vaccine has been recently approved by the U.S. Food and Drug Administration (FDA) and is the primary vaccine during the mpox outbreak in the U.S.

There are certain safety concerns associated with the attenuated virus vaccine ACAM2000, particularly in individuals with immunodeficiency[9–11]. Moreover, the limited availability of the JYNNEOS vaccine underscores the pressing need for a vaccine that can be developed quickly. The use of mRNA vaccines has demonstrated remarkable efficacy in combating *SARS-CoV-2*[12,13]. mRNA vaccines with lipid nanoparticle (LNP) delivery systems have garnered significant attention. These vaccines offer several advantages, including rapid synthesis in

[1]Shanghai Virogin Biotech Co. Ltd., Shanghai, China. [2]Hangzhou Virogin Biotech Co. Ltd., Hangzhou, China. [3]CNBG-Virogin Biotech (Shanghai) Co. Ltd., Shanghai, China. [4]China National Biotec Group Company Limited (CNBG), Beijing, China. [5]Virogin Biotech Canada Ltd., Richmond, Canada. [6]These authors contributed equally: Fujun Hou, Yuntao Zhang, Xiaohu Liu. ✉e-mail: wjia@virogin.com; yangxiaoming@sinopharm.com

cell-free systems, no risk of integrating with the host genome, and the ability to elicit both humoral and cellular immune responses[14].

Extracellular enveloped virus (EEV) and intracellular mature virus (IMV) are the two primary infectious forms of *MPXV*. Subunit vaccines that use selected recombinant viral proteins from EEV and IMV may be safer than live-virus vaccines. Studies have shown that vaccination with the Escherichia coli-expressed A27L, a truncated IMV surface protein, protected mice from a lethal challenge with *VACV*[15]. Additionally, vaccination with recombinant viral EEV proteins B5R or A33R also protected mice from a lethal challenge with *VACV*[16]. In a study by Hooper et al., mice vaccinated with DNA encoding L1R (an IMV protein) and A33R produced neutralizing antibodies against L1R and anti-A33R antibodies. A combination of these two genes was more effective than either gene alone in protecting mice against a lethal challenge with *VACV*[17]. Furthermore, vaccination with DNA encoding VACV genes, including *L1R*, *A27L*, *A33R*, and *B5R*, protected *rhesus macaques* from severe disease following lethal challenge with MPXV[5]. However, Kaufman et al. found that expressing L1R by adenovirus was more effective than the combination of L1R, A27L, A33R, and B5R in protecting mice from a lethal systemic *VACV* infection[18]. For protection against lethal intranasal VACV challenge, however, both L1R and A33R were required[18]. Therefore, further exploration of combination strategies involving potential viral antigens is necessary to enhance vaccine protection against *MPXV*.

*VACV* L1R (also known as L1) is a 250-residue protein located on the surface membrane of IMV[19,20]. Deletion of L1R in *VACV* prevents maturation, and anti-L1R antibodies can block cell entry of IMV, indicating that L1R also plays a role in infection[20–22]. *VACV* A33R is a type II membrane glycoprotein with signal peptide and forms a homodimer outside EEV[23]. A33R is involved in the cell-to-cell spread of EEV[24]. However, there is little information on the homolog of L1R or A33R in *MPXV*. Previous studies have shown that adding a tissue plasminogen activator signal peptide in -frame with L1R results in L1R expression on the cell surface. Notably, surface-exposed L1R induced more anti-L1R neutralizing antibodies[25]. It is important to highlight that antibodies are necessary and sufficient for protecting non-immunized macaques from severe *MPXV*-induced disease[26]. Moreover, optimal protection requires the production of antibodies targeting proteins from both IMV and EEV[4,26].

In this study, we developed mRNA vaccines that encode fusion proteins consisting of EEV and IMV antigens, specifically *MPXV* A35R (the homolog of A33R in *VACV*) and *MPXV* M1R (the homolog of L1R in *VACV*). To enhance the immunogenicity of M1R, we added a signal peptide to A35R extracellular domain to direct the fusion protein. These vaccines were tested for their ability to induce humoral and cellular anti-*VACV* immunity as well as their protection against a lethal dose of *VACV* infection in mice. Our results indicate that the mRNA vaccines outperform the live *VACV* vaccine, as they are able to elicit a robust immune response and provide nearly sterilizing protection against *VACV* challenge.

## Results

### mRNA design and in vitro protein expression

Figure 1a shows that four codon-optimized mRNAs were synthesized for the study, which encode different *MPXV* antigens. The first mRNA encodes full-length wildtype A35R, the second encodes wildtype M1R, the third encodes the integral extracellular domain of A35R fused with full-length M1R (SP-A35R IECD-M1R), and the fourth encodes a shorter extracellular domain of A35R fused with full-length M1R (SP-A35R sECD-M1R). In both fusion proteins, a signal peptide was added to the A35R extracellular domain. The protein expression of the four mRNAs was confirmed by transfecting equal amounts of mRNA into 293 T cells and incubating them with anti-A35R and anti-M1R antibodies, as shown in Fig. 1b. All proteins were expressed and detected by Western blotting, with the mRNA encoding SP-A35R sECD-M1R expressing higher protein level than the mRNA encoding SP-A35R IECD-M1R in 293 T cells.

### A35R- and M1R-specific antibodies and neutralizing antibodies

We have developed three different mRNA-LNP complexes to evaluate their efficacy in inducing protective immunity against *MPXV*. VGPox 1 and VGPox 2 contain LNP-encapsulated SP-A35R IECD-M1R and SP-A35R sECD-M1R, respectively, while VGPox 3 consists of a mixture of two individual mRNA-LNP complexes encoding full-length A35R and M1R, respectively (Fig. 2a). These mRNA vaccines were intramuscularly injected into mice in four groups, as shown in Fig. 2a, and the mice received two doses of vaccination on days 0 and 14 (Fig. 2b). Blood samples were collected on days 7, 13, and 35 for antibody analysis, and the spleens were collected on days 7 and 30 post-vaccination for cellular immunity analysis. On day 36 post the first vaccination, mice were intranasally challenged with a lethal dose of *VACV-WR* ($1 \times 10^6$ PFU/mouse) to evaluate the protective efficacy of the vaccines[27,28].

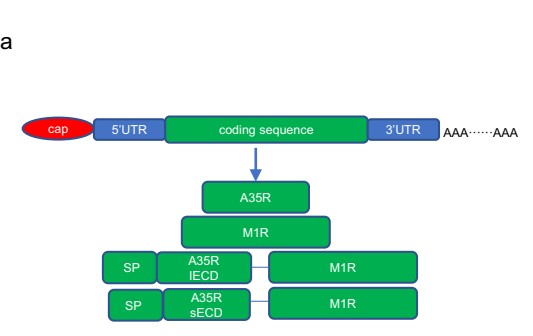

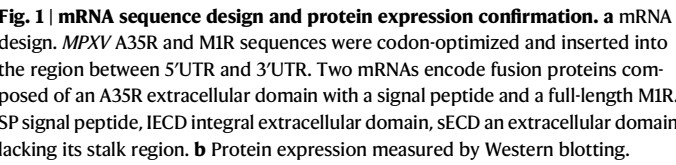

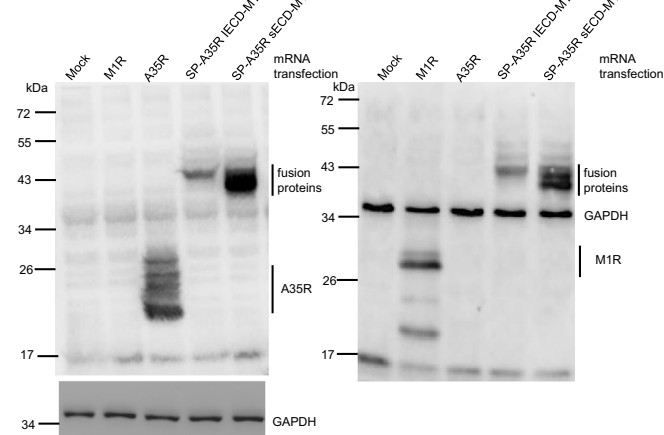

**Fig. 1 | mRNA sequence design and protein expression confirmation. a** mRNA design. *MPXV* A35R and M1R sequences were codon-optimized and inserted into the region between 5′UTR and 3′UTR. Two mRNAs encode fusion proteins composed of an A35R extracellular domain with a signal peptide and a full-length M1R. SP signal peptide, IECD integral extracellular domain, sECD an extracellular domain lacking its stalk region. **b** Protein expression measured by Western blotting.

Each mRNA with 800 ng was transfected into 293 T cells, and then at 16 h post-transfection, the cell lysates were loaded into SDS-PAGE gels for Western blot analysis. Left panel: Western blot analysis with anti-A35R antibody. Right panel: Western blot analysis with anti-M1R and anti-GAPDH antibodies. The immune-blot results were repeated three times.

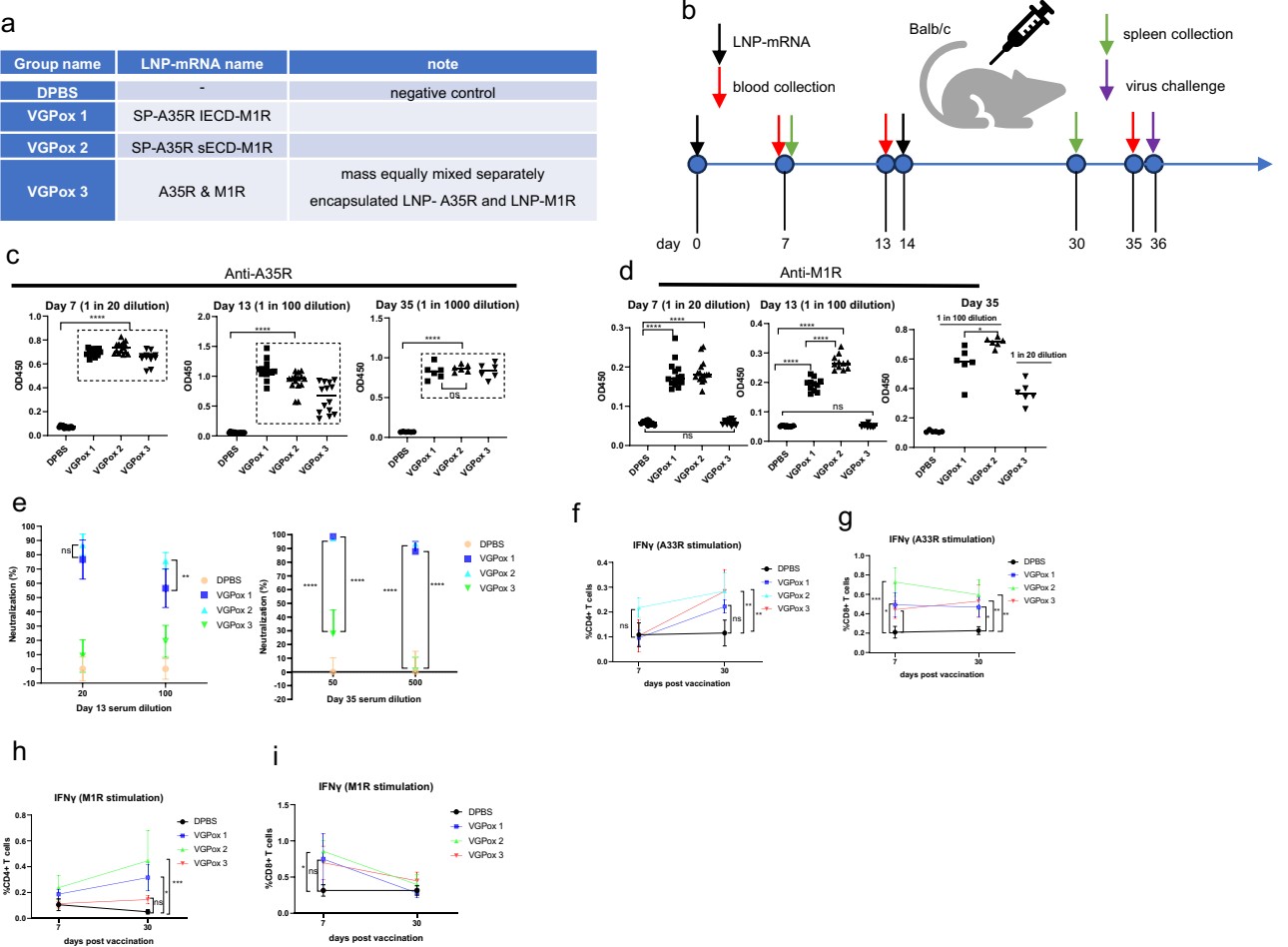

**Fig. 2 | Mice vaccination and immune responses analysis. a** Mice groups. The *Balb/c* mice were randomly divided into four groups, with 20 mice in each group. **b** Mice vaccination and sample collection. The vaccines were administered intramuscularly with a dose of 10 μg on days 0 and 14, respectively. **c** A35R-specific antibodies. The sera collected from mice on days 7, 13, and 35 were tested for A35R-specific antibodies using ELISA against recombinant A35R protein. $n = 12$ to 15 for day 7 and day 13 and $n = 6$ for day 35 biologically independent mice. **d** M1R-specific antibodies. The sera were tested for M1R-specific antibodies using ELISA against recombinant M1R protein. $n = 12$ to 15 for day 7 and day 13 and $n = 6$ for day 35 biologically independent mice. **e** Neutralizing antibodies. The sera were tested for neutralizing antibodies against *VACV* using plaque reduction neutralization test (PRNT). $n = 6$ biologically independent mice. Data are presented as mean values ± SD.

**f** CD4⁺ T cell immune response against A35R. The splenocytes were analyzed for the percentage of CD4⁺ T cells producing IFN-γ in response to A33R (VACV homolog of A35R) stimulation. $n = 4$ for day 7 and $n = 5$ for day 30 biologically independent mice. Data are presented as mean values ± SD. **g** CD8⁺ T cell immune response against A35R. $n = 4$ for day 7 and $n = 5$ for day 30 biologically independent mice. Data are presented as mean values ± SD. **h** CD4⁺ T cell immune response against M1R. $n = 4$ for day 7 and $n = 5$ for day 30 biologically independent mice. Data are presented as mean values ± SD. **i** CD8⁺ T cell immune response against M1R. $n = 4$ for day 7 and $n = 5$ for day 30 biologically independent mice. Data are presented as mean values ± SD. All the data in (**c**–**i**) were analyzed by one-way ANOVA with Tukey's multiple comparison tests. "ns" indicates "not significant"; "*" indicates $p < 0.05$; "**" indicates $p < 0.01$; "***" indicates $p < 0.001$; "****" indicates $p < 0.0001$.

All three vaccine groups induced anti-A35R antibodies by day 7 post vaccination, with titers increasing on days 13 and 35 (Fig. 2c). However, anti-M1R antibodies were only detected in VGPox 1 and VGPox 2 groups on days 7 and 13, and low titers were observed in the VGPox 3 group only on day 35 (Fig. 2d). Moreover, neutralizing activity against live virus (*VACV-WR*) was observed in sera from VGPox 1 and VGPox 2 groups on day 13, but not in the VGPox 3 group (Fig. 2e, left panel). By day 35, sera from all three mRNA vaccines showed partial or nearly complete neutralization of the virus at a 1 in 50 dilution. Notably, VGPox 1 and VGPox 2 demonstrated significantly higher neutralizing ability than VGPox 3 at a 1 in 500 dilution (Fig. 2e, right panel).

## mRNA vaccines can activate T cell immune response

To investigate the cellular immunity induced by the mRNA vaccines, we isolated cells from the spleens of vaccinated mice on day 7 and day 30 post-vaccination and measured elevated IFN-γ in CD4⁺ and CD8⁺ T cells using flow cytometry analysis. To stimulate CD4⁺ and CD8⁺ T cells for *MPXV* A35R, we used its homolog vaccinia A33R protein. The

percentage of A33R-specific CD8⁺ T cells was increased even at day 7 post-vaccination with all three vaccines (Fig. 2g). However, the A33R-specific CD4⁺ T cells only increased at day 30 (Fig. 2f). On day 30 post-vaccination, VGPox 1 and VGPox 2 but not VGPox 3 induced M1R-specific CD4⁺ T cells (Fig. 2h). These results suggest that the mRNA vaccines can activate T cell immune response, which may contribute to the protective immunity against the virus. In general, At the later time point, the vaccinated groups exhibited a higher level of activated antigen-specific CD4⁺ T cells. In contrast, the antigen-specific CD8⁺ T cells were present at the early time point, but their percentages either remained unchanged or declined at the late time point (Fig. 2g, i).

## All three vaccines are protective in mice challenged with a lethal dose virus

To evaluate the protective efficacy of the mRNA vaccines, the mice were challenged with a lethal dose of *vaccinia virus* (*VACV-WR*) via the intranasal route. The DPBS control group showed a significant

decrease in body weight, dropping to almost 70% of their initial weight by day 8 and 2 of 5 mice died on day 9 and 11 post virus challenge. In contrast, mice in all three mRNA vaccine groups showed no significant body weight loss or any other abnormalities (Fig. 3a). The complete virus clearance was observed in the lungs of the vaccinated mice on 9 days post-infection, whereas the control group exhibited a high viral load in the lungs (Fig. 3b). These results suggest that the mRNA vaccines effectively protected the mice from the *VACV-WR* challenge.

### The weak immunogenicity of M1R in VGPox 3 is independent of A35R

VGPox 1 and VGPox 2, which encode the fusion antigens of A35R and M1R, induced higher levels of antibodies against M1R and better neutralizing ability than VGPox 3, a mixture of A35R-LNP and M1R-LNP complexes. To rule out the possibility of A35R interfering with M1R immunity when the two mRNAs are co-expressed in the same cells, we vaccinated animals with M1R mRNA-LNP or A35R mRNA-LNP alone. As shown in Fig. 4a, b, A35R antibodies were readily detectable on day 7 post-vaccination, but M1R mRNA-LNP failed to induce M1R antibodies. Once again, VGPox 1–3 induced A35R antibodies, but only VGPox 1 and 2 were able to induce anti-M1R antibodies at this time point.

### A single dose of mRNA vaccine protected mice from *VACV* challenge after 7 days of immunization

To further evaluate the speed of the mRNA vaccines in inducing protective immunity, we implemented a single-dose 8-day vaccination regimen followed by *VACV-WR* challenge (Fig. 5a). Additionally, we intraperitoneally vaccinated mice with *VACV-WR* as a live-virus vaccine at two sublethal doses (*VACV*-low and *VACV*-high). As shown in Fig. 5b, all three mRNA vaccines induced corresponding antibodies against A35R. Consistent with previous findings (Fig. 2d), only VGPox 1 and VGPox 2 were able to elicit antibodies against M1R (Fig. 5c), while vaccination with sublethal live *VACV-WR* failed to produce antibodies against the two antigens on day 7 (Fig. 5b, c). Interestingly, sera from animals vaccinated with VGPox 1 and 2 showed a strong ability to neutralize live virus by 87.8% and 93.4%, respectively, indicating their potential as effective vaccines (Fig. 5d). However, Live-virus vaccination exhibited only approximately 50% neutralization ability, whereas VGPox 3-vaccinated sera were even less effective, with only around 20% ability to neutralize the virus at this time point (Fig. 5d). As shown in Fig. 5e, all three mRNA candidates and *VACV*-high provided complete protection, with 100% survival rates. *VACV*-low exhibited slightly inferior protection compared to the others.

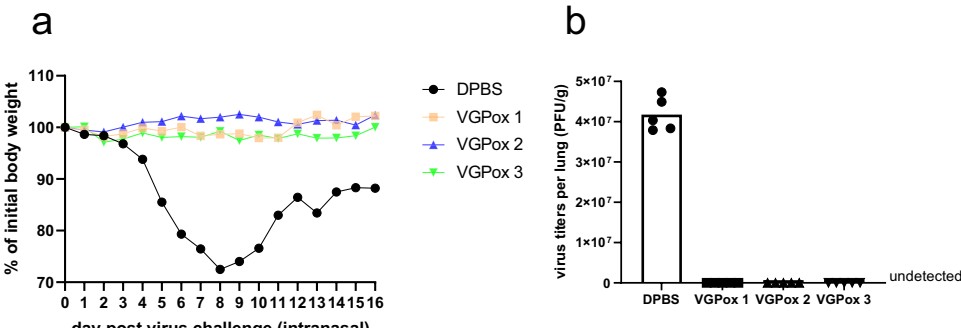

**Fig. 3 | Protection of mRNA vaccination for virus challenge. a** Virus-challenge protection. At day 36, mice were intranasally challenged with a lethal dose (1 × 10⁶ PFU) of WR vaccinia virus. The body weight of each mouse was examined daily post virus challenge, and the changes were calculated and compared with the initial weight. *n* = 5 biologically independent mice. **b** Viral load in the lungs. The lungs were collected on day 4 in the DPBS group or on day 9 post virus inoculation in other groups. Virus titers were examined by plaque assay. *n* = 5 biologically independent mice.

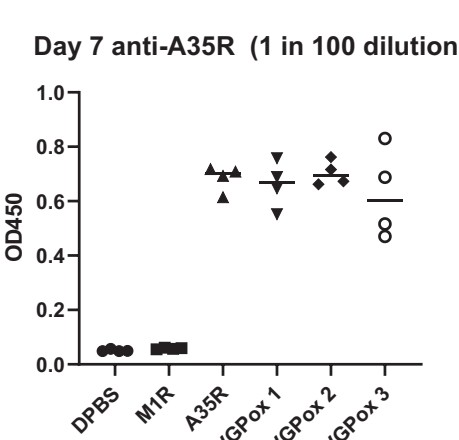

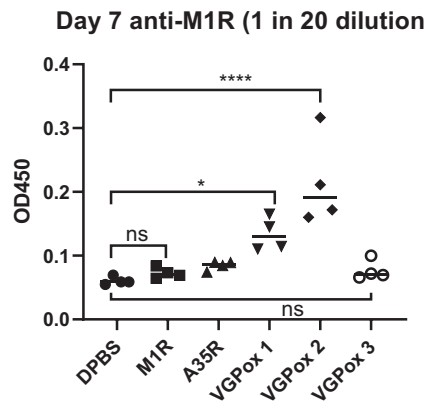

**Fig. 4 | The weak immunogenicity of M1R in VGPox 3 is independent of A35R.** *n* = 4 biologically independent mice. To evaluate the antigen-specific antibody response, mice were divided into six groups, each consisting of four mice. Each mouse was vaccinated with 10 μg of LNP-encapsulated VGPox 1, VGPox 2, VGPox 3, M1R alone, or A35R alone via intramuscular injection. Blood samples were collected from each mouse on day 7 post-vaccination, and antigen-specific antibodies for A35R (**a**) and M1R (**b**) were analyzed using ELISA. All the data were analyzed by one-way ANOVA with Tukey's multiple comparison test. "ns" indicates "not significant"; "*" indicates *p* < 0.05; "****" indicates *p* < 0.0001.

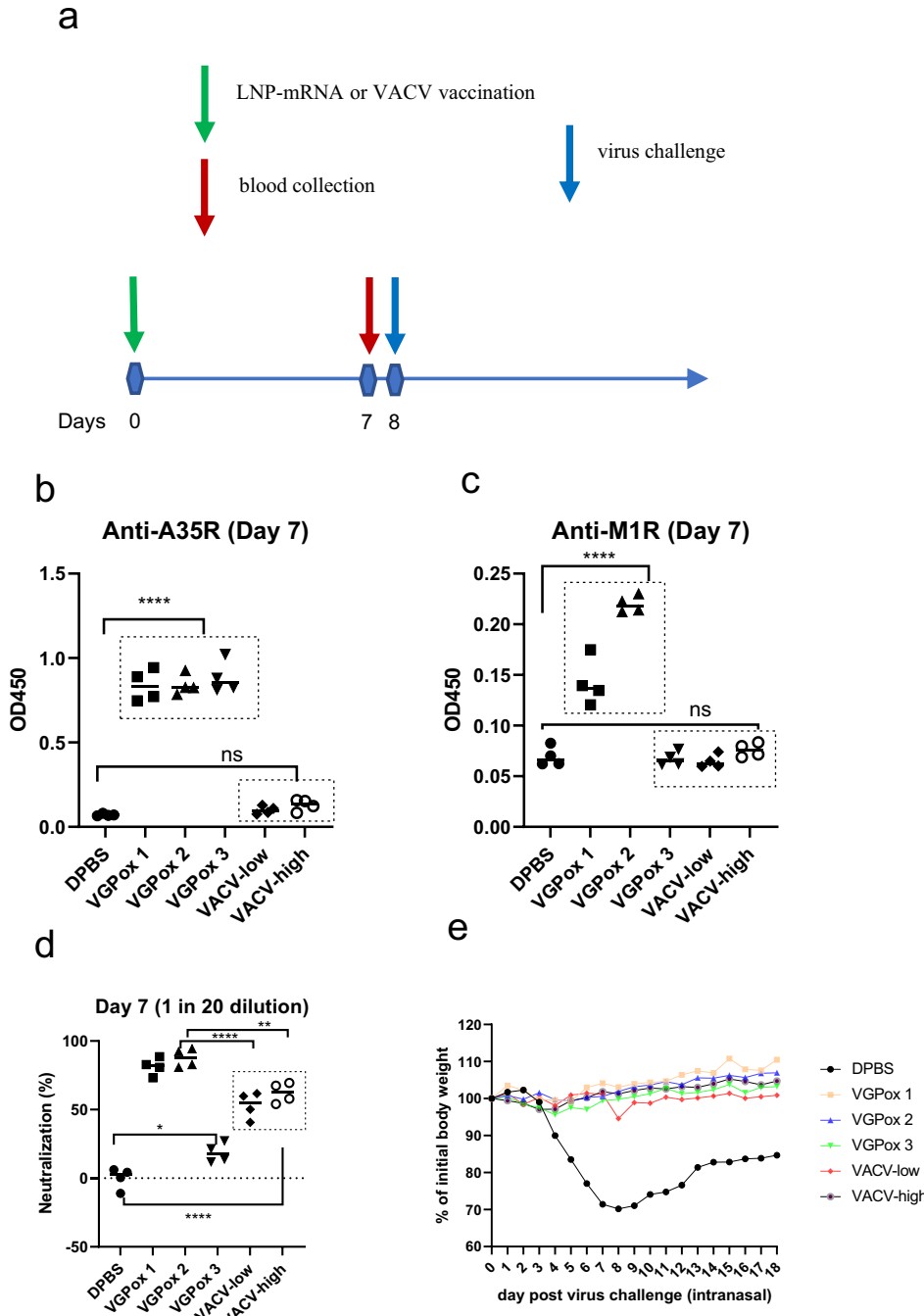

**Fig. 5 | Single-dose vaccination, immune responses, and virus challenge. a** Mice were divided into 6 groups: 3 mRNA vaccine groups (VGPox 1, VGPox 2, VGPox 3), a low-dose live *VACV-WR* group ($2 \times 10^4$ PFU/dose), a high-dose live *VACV-WR* group ($2 \times 10^5$ PFU/dose), and a negative control group (DPBS). Each mouse was vaccinated with 10 μg/dose of mRNA or live *VACV-WR* at day 0. **b** Anti-A35R and **c** anti-M1R antibody levels were analyzed by ELISA. All serum samples were diluted 1 in 20. *n* = 4 biologically independent mice. **d** The neutralizing activity of sera collected on day 7 post-vaccination was measured by *VACV* PRNT. Sera from the DPBS group

were considered as 0% neutralization. n = 4 biologically independent mice. All the data in (**b**–**d**) were analyzed by one-way ANOVA with Tukey's multiple comparison tests. "ns" indicates "not significant"; "*" indicates $p < 0.05$; "**" indicates $p < 0.01$; "***" indicates $p < 0.001$; "****" indicates $p < 0.0001$. **e** On day 8 post-vaccination, mice from each group were intranasally challenged with *VACV-WR* ($1 \times 10^6$ PFU/mouse in 20 μl), and their body weight was recorded daily to assess disease severity. *n* = 5 biologically independent mice.

## Long-term protection of the mRNA vaccines

To evaluate the long-term protective immunity of the mRNA vaccines, mice were vaccinated with two doses of VGPox 1–3. The A35R-specific antibodies gradually decreased from the 1st until the 5th month for all three vaccine candidates (Fig. 6a). In contrast, the levels of M1R-specific antibodies slightly increased at the 2nd month and then

decreased at the 3rd month for VGPox 1 and 2, while for VGPox 3, it induced relatively stable low-level anti-M1R antibodies within 5 months, similar to the live virus *VACV* vaccination group (Fig. 6b). Consistent with previous results, VGPox 1 and 2 induced higher levels of neutralizing antibodies than VGPox 3 or live *VACV* vaccination (Fig. 6c).

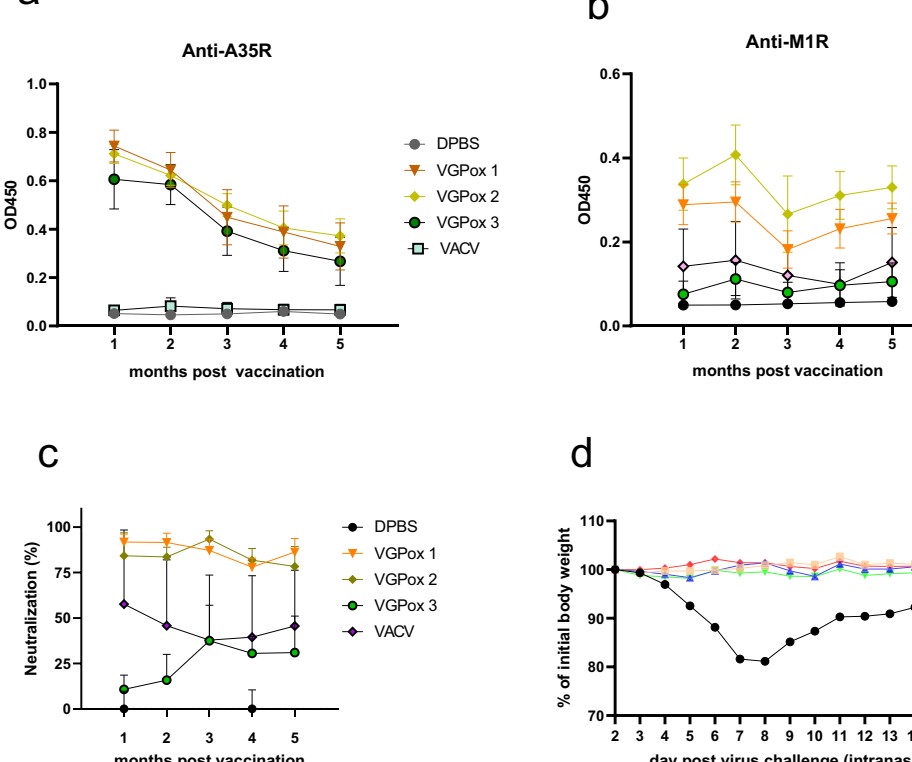

**Fig. 6 | Long-term immunity test.** Mice were divided into four groups and intramuscularly vaccinated with VGPox1, VGPox2, VGPox3 (10 µg/dose, twice on day 0 and day 14), or intraperitoneally injected with live virus *VACV-WR* (2 × 10⁵ PFU/dose). Blood was collected monthly after the first dose vaccination for analysis of anti-A35R (**a**), anti-M1R (**b**) using ELISA as described previously and neutralizing antibodies (**c**). The sera from each mouse were 1 in 500 diluted and incubated with

*VACV-WR* virus for *VACV* PRNT. Sera from the DPBS group were regarded as 0% neutralization. *n* = 8 biologically independent mice. Data are presented as mean values ± SD. After 5 months, mice were challenged intranasally with 1 × 10⁶ PFU *VACV-WR*, and their body weight was recorded daily (**d**). *n* = 8 biologically independent mice.

On 162 days post-vaccination, mice were intranasally challenged with lethal *VACV-WR*, and all three mRNA candidates and live virus vaccination provided complete protection against the *VACV* challenge (Fig. 6d). These results suggest that the mRNA vaccines can induce long-term protective immunity against *VACV* and that VGPox 1 and 2 are more effective than VGPox 3 in inducing neutralizing antibodies.

**Passive protection assay showed VGPox 2 provided better protection**

To determine the protective efficacy of the sera obtained from mice vaccinated with the mRNA candidates, naïve mice were administered with the sera obtained from the animals vaccinated five months earlier (as shown in Fig. 6). The following day, the mice were challenged with *VACV-WR* and monitored for body weight changes. Figure 7a shows that most animals in the DPBS, VGPox 1, VGPox 3, and *VACV* groups experienced significant body weight loss after the virus challenge, with two mice in the DPBS group dying on day 9 post-challenge. In contrast, three out of four mice in the VGPox 2 group had either no or minor changes in their body weight, as shown in Fig. 7d, suggesting that the sera obtained from VGPox 2-immunized mice may confer better protection.

## Discussion

The current study evaluated the efficacy of three mRNA vaccine candidates for poxvirus. VGPox1 and VGPox2 are single mRNA molecules that encode a fusion protein comprising the extracellular domain of A35R with a signal peptide and a full-length M1R, while VGPox3

consists of a mixture of two individual mRNA–LNP complexes that encode the wild-type A35R and M1R, respectively.

The key findings of the current study reveal that mRNA vaccines encoding the fusion forms of A35R and M1R (VGPox 1 and VGPox 2) can efficiently stimulate high levels of both A35R and M1R antibodies and effectively neutralize live-virus infections in cell cultures. Conversely, VGPox 3, which is a mixture of two individual mRNAs encoding the wildtype A35R and M1R, did not exhibit similar results. Although VGPox 3 induced comparable levels of A35R antibodies, M1R-specific antibodies appeared much later. Notably, the sera collected from VGPox 3-vaccinated animals at early time points were unable to neutralize the virus, highlighting the importance of anti-M1R antibodies for the neutralization of the virus in vitro.

However, it should be noted that all three mRNA vaccine candidates were able to provide complete protection in animals challenged with the live virus as early as 7 days post-vaccination, regardless of the levels of anti-M1R-specific antibodies. Therefore, although the role of anti-M1R immunity against intranasal *VACV* infection is still unclear, the current study suggests that the vaccine-induced protection against *VACV* may involve other mechanisms in addition to the induction of M1R-specific antibodies.

In contrast to VGPox1 and 2, which both encode fusion proteins of A35R and M1R, VGPox 3 encodes the individual mRNAs for A35R and M1R. Our results show that VGPox 3 was not able to induce significant levels of M1R-specific antibodies within 7 days post-vaccination (Fig. 4b). These findings suggest that the fusion of M1R to A35R with a signal peptide is crucial for enhancing the immunogenicity of M1R

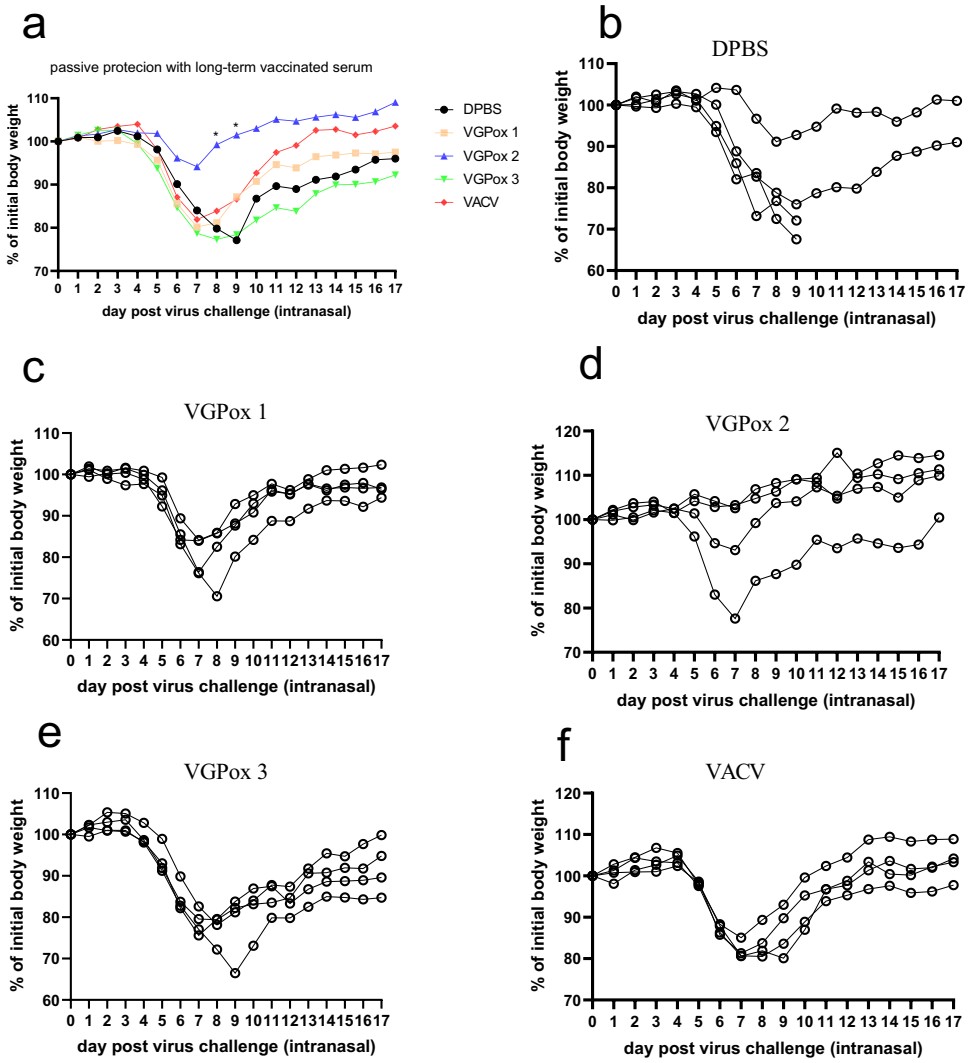

**Fig. 7 | Passive protection by the long-term immunity sera.** Naïve mice were intravenously injected with 100 μL of sera obtained from the long-term immunity group, which contained an equal-volume mixture of sera collected from months 1–4 post vaccination. The following day, the mice were intranasally challenged with $1 \times 10^5$ PFU *VACV-WR*. The mean body weight change post virus infection (**a**) and body weight changes for individual animals were recorded and analyzed for DPBS (**b**), VGPox 1 (**c**), VGPox 2 (**d**), VGPox 3 (**e**), and *VACV* (**f**). $n = 4$ biologically independent mice. Comparison between DPBS and VGPox2 on day 8 and day 9 were analyzed by two-tailed *t*-test. "*" indicates $p < 0.05$.

and creating a bivalent vaccine that provides potent immunity against both antigens, as demonstrated by the efficacy of VGPox1 and 2.

VGPox 1 and VGPox 2 induce earlier production of neutralizing antibodies against A35R and M1R compared to the sublethal *VACV-WR* virus. Surprisingly, live-virus-vaccinated animals did not produce anti-A35R and anti-M1R antibodies on day 7 post-vaccination, yet their sera could still neutralize the live virus, albeit with ~50% of the potency observed in animals vaccinated with VGPox 1 and 2. This suggests that the live virus may induce humoral immunity against other viral antigens soon after vaccination, which is also effective in protecting against virus infection. Previous research by Hooper et al. has proposed that viral proteins A27L and B5R could be among the possible antigens responsible for this effect[5].

The mRNA vaccines can induce T-cell immunity against both A35R and M1R. However, the time course for developing antigen-specific CD4[+] and CD8[+] induced by VGPox 1–3 is intriguingly opposite, with an increase in CD4[+] but a decrease or no change in CD8[+] observed on day 30 post-vaccination as compared to day 7. Although increased numbers of antigen-specific CD8[+] T cells were observed in the early time point post-vaccination, the increased antigen-specific CD4[+] T cells

mainly appeared in the late time point. Similar results were also found in mice spleen samples extracted from day 7 post *VACV* infection[29]. In contrast, the *SARS-CoV-2* mRNA vaccine elicited a rapid response in CD4[+] T cells rather than CD8[+] T cells at early time points[30]. Further investigation into the types of CD4[+] T cells and their potential role in the development of M1R humoral immunity at the late time point would be of interest. However, it should be noted that previous studies have suggested that cellular immunity may not play a crucial role in protection against *VACV* in both mice and non-human primates[4,26]. This is supported by the fact that passive transfer of vaccinated serum has been shown to provide adequate protection in mice and macaques in previous studies[4,26]. Our current results also support this finding, as sera obtained from animals vaccinated with VGPox 2 were able to confer sufficient protection against virus challenge in naïve mice (Fig. 7a).

In conclusion, our study demonstrated the superiority of mRNA vaccines expressing fusion proteins composed of *MPXV* A35R extracellular domain with a signal peptide and M1R over the sublethal live *VACV-WR* virus in terms of anti-virus immunity. The early induction of humoral immunity against the virus by VGPox 1 and VGPox 2 as early as

7 days post-vaccination resulted in complete protection against the lethal vaccinia virus challenge. Our findings are particularly noteworthy because of the high homology between vaccinia and mpox, suggesting that VGPox 1 or 2 could serve as promising mRNA vaccines against other orthopoxviruses. While this study provides valuable insights into the efficacy of mRNA vaccines against mpox, there are some limitations that should be acknowledged. For instance, the study only used *VACV* but not *MPXV* to conduct in vitro neutralization and in vivo experiments. Future studies could expand upon these findings by comparing the mRNA vaccine to other approved vaccines, such as JYNNEOS, in non-human primates. In addition, the authors plan to conduct further experiments to address these limitations and gain a more comprehensive understanding of the potential of mRNA vaccines against mpox.

## Methods

### Cell lines and virus

The Vero cells and 293 T cells used in this study were originally obtained from American Type Culture Collection (ATCC) and cultured in complete Dulbecco's Modified Eagle Medium (DMEM) supplemented with 10% fetal bovine serum (FBS) and 1% penicillin-streptomycin. The *Vaccinia virus Western Reserve* (WR, VR-1354) was also obtained from ATCC and propagated in Vero cells. Infected cell lysates and supernatants were collected and ultracentrifuged, and virus titers were quantified using routine plaque assay.

### Antigen sequences and plasmids

M1R and A35R protein sequences from *MPXV* strain Zaire79 were reverse-translated into their coding sequences using GenSmart™ Codon Optimization (GenScript). The coding sequences were then synthesized by Azenda Life Sciences and cloned into pUC vectors containing an upstream T7 RNA polymerase promoter and downstream polyA sequences. The A35R extracellular domain with a signal peptide was fused with M1R by a peptide linker. The accuracy of the sequences was confirmed by DNA sequencing.

### In vitro transcription

The corresponding mRNAs were prepared by Fraserna Life Sciences Limited. Briefly, Plasmids were first linearized by the enzyme BSPQI (Vazyme, Nanjing) to serve as a template for mRNA preparation. The next step involved incubating the linearized plasmids with T7 RNA polymerase, nucleoside triphosphates (NTPs), and RNase inhibitors at 37 °C for 2 h. The mRNA was then purified using a commercial kit, and the integrity of the mRNA was confirmed by agarose gel analysis.

### Transfection and Western blot

mRNA transfection was carried out according to the Lipofectamine 3000 protocol with slight modification. 293 T cells were seeded in 24-well plates and transfected with 800 ng mRNA per well. At 16 h post-transfection, cells were collected by scraping in SDS loading buffer (Beyotime, Shanghai) and boiled at 95 °C for 5 min. The antibodies used in this study were as follows: M1R Human Mab (OkayBio, R403k5), 1:1000; A35R Mouse Mab (Sino Biological, 40886-M0017), 1:1000; and GAPDH Mouse Mab (Sangon, D190090), 1:2000. HRP-goat anti-human IgG (Sangon, D110150), 1:5000, and HRP-goat anti-mouse IgG (Sangon, D110087), 1:5000.

### LNP encapsulation

mRNA was encapsulated within an LNP formulation by Fraserna Life Sciences Limited using a Nanoassemblr® microfluidic device (Precision Nanosystems Inc., Vancouver, Canada), as described in previous studies[31,32]. The raw mRNA-LNP was then subjected to dialysis to remove residual citrate salt and ethanol. Finally, the dialyzed mRNA–LNP solution was stored at −80 °C with a cryoprotectant to maintain stability.

### Mice vaccination

*Balb/c* mice (female, 7–8 weeks old) were obtained from Vitalriver (Beijing). Mice were cultured at ambient temperature of 20–26 °C and humidity of 40% ∼ 70% with 12 h/12 h of dark/light. Ten micrograms of mRNA-LNP in 100 μL PBS were intramuscularly injected per mouse. Some mice were boosted with the same dose 14 days post first vaccination. In other experiments, mice were vaccinated (intraperitoneally [IP]) with sublethal doses of *VACV-WR* ($2 \times 10^4$ PFU or $2 \times 10^5$ PFU) as a positive control.

### Enzyme-linked immunosorbent assay

To measure the antibody response, A35R protein (OkayBio, C1620) and M1R protein (OkayBio, C1624) were diluted to 5 μg/mL using enzyme-linked immunosorbent assay coating buffer (Elabscience, E-ELIR-003) and added to 96-well plates for overnight coating at 4 °C. The plates were washed with washing buffer to remove any uncoated proteins before incubating with blocking buffer overnight. Sera were collected from different time points and centrifuged before being added to the wells at different dilutions. After incubation, the plates were washed and incubated with HRP-goat anti-mouse IgG (Sangon, D110087), followed by washing steps and final optical density analysis in an MD microplate reader to determine the antibody response.

### Plaque reduction neutralization test

The plaque reduction neutralization test was performed to assess the neutralizing antibodies present in the serum samples. Initially, the sera were heat-inactivated at 56 °C for 30 min. Subsequently, 100 μL of sera with varying dilutions was added to 200 PFU of virus (*VACV-WR*) in a 96-well plate and incubated at 37 °C for 1 h. The virus-sera mixture was then added to Vero cells and incubated for 1 h at 37 °C while shaking every 15 min to ensure complete access of the virus to the cells. Following this, 500 μL of 1% methylcellulose was added to each well, and the cells were incubated at 37 °C for 2 days to allow for plaque formation. The cells were then fixed and stained with 1% crystal violet. The number of plaques was counted to determine the neutralizing antibody titers in the sera samples.

### Flow cytometric analysis

Spleens were harvested from vaccinated mice on days 7 and 30 post first immunization. The spleens were mechanically disrupted and treated with ACK lysis buffer to remove red blood cells. Spleen cells were filtered through a 70 μm cell strainer and resuspended in RPMI1640 medium. Subsequently, $1 \times 10^6$ spleen cells were plated in a 96-well plate and stimulated with the indicated protein for 24 h. Golgiplug was added for 4 h before staining to inhibit protein transport. The cells were then stained with antibodies targeting surface markers for 30 min on ice, followed by fixation and permeabilization for another 30 min. For intracellular staining, fixed cells were washed and stained with intracellular antibodies for 1 h on ice. Stained cells were analyzed with a Cytoflex flow cytometer (Beckman) and FlowJo V.10.8.1 software. The gating strategy is shown in Supplementary Fig. 1. Listed antibodies: Alexa Fluor® 700 anti-mouse CD45 Antibody, 147716, Biolegend; FITC anti-mouse CD3ε Antibody, 100306, Biolegend; Brilliant Violet 785™ anti-mouse CD4 Antibody, 100552, Biolegend; Brilliant Violet 605™ anti-mouse CD8a Antibody, 100744, Biolegend; Zombie NIR™ Fixable Viability Kit, 423106, Biolegend; CD69 Monoclonal Antibody (H1.2F3), APC, eBioscience™, 17-0691-82, Invitrogen; PE anti-mouse IFN-γ Antibody, 505808, Biolegend; PerCP/Cyanine5.5, anti-mouse IL-2 Antibody, 503822, Biolegend; PE/Cyanine7 anti-mouse TNF-α Antibody, 506324, Biolegend; Brilliant Violet 421™ anti-mouse/human IL-5 Antibody, 504311, Biolegend.

### Virus challenge

After anesthesia, $1 \times 10^6$ PFU *VACV-WR* virus in 20 μL was intranasally administered to the mice. Mice were monitored daily for changes in

body weight and clinical symptoms and were euthanized either at the end of the experiment or when weight loss exceeded 30% of their initial weight. Lungs were collected and preserved in complete DMEM for further analysis. For the passive protection assay, a challenge dose of $1 \times 10^5$ PFU *VACV-WR* by nasal inoculation was used, which is approximately the LD50 of the virus in Balb/c mice when infected via this route. All experimental animal procedures were approved by the Animal Care Committee of Shanghai Virogin Biotech Co., Ltd. (permission number for each experiment: RD-VC-2022003; RD-VC-2022007; RD-VC-2022015; RD-VC-2023006; RD-VC-2023007; RD-VC-2023011; RD-VC-2023012).

### Viral load in lungs
Lungs were ground in a tissue homogenizer followed by three freeze–thaw cycles to release the virus from cells. After centrifugation, supernatants at different dilutions were added to Vero cells for a plaque assay.

### Statistical analysis
Data analysis was performed using Microsoft Excel 2019 and GraphPad Prism 9 software, and one-way ANOVA was used to analyze differences in multiple groups where "*" indicates $p < 0.05$, "**" indicates $p < 0.01$, "***" indicates $p < 0.001$, and "****" indicates $p < 0.0001$. A two-tailed $t$-test was used to compare data from the two groups.

### Reporting summary
Further information on research design is available in the Nature Portfolio Reporting Summary linked to this article.

## Data availability
Datasets generated and/or analyzed during the current study are available within the paper or are appended as supplementary data. Source data are provided in this paper.

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

## Acknowledgements
We would like to thank Min Liang for the discussion of poxvirus information.

## Author contributions
X.Y., W.J., H.H., R.Z., Y.M., F.H., and X.L. designed the study. F.H., Y.Z., J.X., Z.Y., J.D., X.H., and Y.S. performed the experiments. F.H., Z.Y., J.X., and W.J. wrote the paper.

## Competing interests
X.Y. and Y.Z. are employees of CNBG or its subsidiary companies. The rest of the authors are employees of Virogin. All the above companies are involved in the development of mRNA vaccines.
