## [Peer Review File · Nature Communications]

REVIEWER COMMENTS

Reviewer #1 (Remarks to the Author):

Hou and colleagues have shown that mRNA vaccines can protect mice from a lethal virus challenge. Overall, the data is of interest and offers a possible alternative to current pox virus vaccine strategies.

The passive protection study in Fig. 6B shows essentially no protection. The authors should discuss this.

There are numerous awkward sentences. The manuscript should be carefully edited by a native English speaker.

Line 35 and throughout – Please use mpox

Line 51 – “Since the overwhelming success of the mRNA vaccine in defeating 51 SARS-CoV-2...” this is over-stated. Please revise.

Line 61 – “Hooper et al. found that mice vaccination with DNA encoding 61 L1R (an IMV protein) and A33R induced neutralizing antibodies against L1R and IgGs against 62 A33R.” Please revise.

Line 69 – “Therefore, selection of viral antigens and combination strategies for better vaccine protection 69 against the MPXV remain to be explored.” Selection has certainly been explored, but the optimum combination not yet defined. Please revise.

Line 84 – “These vaccines were tested for their humoral and cellular anti-VACV immunities as well as their protection against a lethal dose of VACV infection in mice.” Vaccines were tested for ability to induce cellular and humoral immunity. Not as stated. Please revise.

Line 143 - In general, both antigens were able to stimulate more CD4+ T cells in the late time point compared 143 to the early time point.” This is a misleading. The stimulation was earlier. Please revise.

Line 144 Replace “Contrarily” with “In contrast.”

Line 174 – “However, live-virus vaccination only showed partial neutralization ability...” Please revise for clarity.

Line 237 “However, the time course for the development of antigen-specific CD4+ and CD8+ induced by VGPOx 1 - 3 seems opposite.” Please revise for clarity.

Issues with Figures.

Fig. 1B – Composite image should be clearly indicated with a line between spliced lanes.

Figs. 2, 5 and 7 - Replaces IgGs – this is not an accurate description of the studies.

Reviewer #2 (Remarks to the Author):

The paper describes some promising preliminary results for a poxvirus vaccine. An mRNA vaccine for orthopoxviruses, particularly mpox, could be potentially useful as these vaccines are easier to ramp up production than live vaccines. There are some errors related to citations, and one of the figure legends that should be corrected. The in vivo studies could have been more robustly executed. Additionally, although the work has promising preliminary results, there are some definite limitations and additional work that should be done in the future. These limitations should be included in the discussion. Please see the attached PDF with my detailed comments.

Reviewer #1 (Remarks to the Author):

Hou and colleagues have shown that mRNA vaccines can protect mice from a lethal virus challenge. Overall, the data is of interest and offers a possible alternative to current pox virus vaccine strategies. Thank you for your review and valuable comments. Below are the point-by-point responses and the corresponding corrections are highlighted in yellow in the revised manuscript.

The passive protection study in Fig. 6B shows essentially no protection. The authors should discuss this.

We replicated the experiment and found consistent results with our previous study, suggesting that sera collected 7 days after immunization did not provide protection against virus challenge. However, we observed a marked improvement in protection when mice received intravenous injections of sera collected from long-term immunized mice vaccinated with VGPOX 2, as shown in Figure 6. These findings lead us to hypothesize that passive protection requires a certain quantity of immunized sera, which cannot be achieved with the 7-day sera, but can be achieved with long-term immunization. We have replaced the passive immunity results with the new one (Figure 6).

There are numerous awkward sentences. The manuscript should be carefully edited by a native English speaker.

The manuscript has now undergone extensive revisions to enhance the clarity and coherence of the English language.

Line 35 and throughout – Please use mpox

All “monkeypox” have been replaced with “mpox”.

Line 50 – “Since the overwhelming success of the mRNA vaccine in defeating SARS-CoV-2...” this is over-stated. Please revise.

It has been changed to “The use of mRNA vaccines has demonstrated remarkable efficacy in combating SARS-CoV-2”.

Line 62 – “Hooper et al. found that mice vaccination with DNA encoding L1R (an IMV protein) and A33R induced neutralizing antibodies against L1R and IgGs against A33R.” Please revise.

Yes, “IgGs against A33R” has been changed to “anti-A33R antibodies”.

Line 69 – “Therefore, selection of viral antigens and combination strategies for better vaccine protection against the MPXV remain to be explored.” Selection has certainly been explored, but the optimum combination not yet defined. Please revise.

It has been changed to “further exploration of combination strategies involving potential viral antigens is necessary to enhance vaccine protection against MPXV.”

Line 85 – “These vaccines were tested for their humoral and cellular anti-VACV immunities as well as their protection against a lethal dose of VACV infection in mice.” Vaccines were tested for ability to induce cellular and humoral immunity. Not as stated. Please revise.

It has been revised to “These vaccines were tested for their ability to induce humoral and cellular

anti-VACV immunity as well as their protection against a lethal dose of VACV infection in mice.”

Line 162 - In general, both antigens were able to stimulate more CD4+ T cells in the late time point compared to the early time point.” This is a misleading. The stimulation was earlier. Please revise. It has been revised to “At the later time point, the vaccinated groups exhibited a higher level of activated antigen-specific CD4+ T cells.”

Line 162 Replace “Contrarily” with “In contrast.”

It has been revised.

Line 200 – “However, live-virus vaccination only showed partial neutralization ability...” Please revise for clarity.

It has been changed to “Live-virus vaccination exhibited only approximately 50% neutralization ability, whereas VGPOx 3-vaccinated sera were even less effective, with only around 20% ability to neutralize the virus at this time point.”

Line 288 “However, the time course for the development of antigen-specific CD4+ and CD8+ induced by VGPOx 1 - 3 seems opposite.” Please revise for clarity.

It has been revised to “the time course for developing antigen-specific CD4+ and CD8+ induced by VGPOx 1-3 is intriguingly opposite, with an increase in CD4+ but a decrease or no change in CD8+ observed on day 30 post-vaccination as compared to day 7.”

Issues with Figures.

Fig. 1B – Composite image should be clearly indicated with a line between spliced lanes.

It has been revised as suggested.

Figs. 2, 5 and 7 - Replaces IgGs – this is not an accurate description of the studies.

Thanks for your reminder that the used secondary antibodies in ELISA can also react light chain of other immunoglobulins besides IgGs. All the related figures have been revised.

Reviewer #2 (Remarks to the Author):

The paper describes some promising preliminary results for a poxvirus vaccine. An mRNA vaccine for orthopoxviruses, particularly mpox, could be potentially useful as these vaccines are easier to ramp up production than live vaccines. There are some errors related to citations, and one of the figure legends that should be corrected. The in vivo studies could have been more robustly executed. Additionally, although the work has promising preliminary results, there are some definite limitations and additional work that should be done in the future. These limitations should be included in the discussion. Please see the attached PDF with my detailed comments.

Thank you for reviewing our manuscript and providing valuable feedback. We have taken your comments into account and repeated all in vivo experiments, with the observation period extended to at least two weeks. Furthermore, we have included the results of a long-term vaccination experiment that was initiated before manuscript submission in Figure 5. Please find our point-by-point responses to your comments below, along with the revised portions highlighted in green within

the revised manuscript.

Line 16 JYNNEOS is considered attenuated live non-replicating vaccine, please add for clarity as it does have a favorable safety profile.

Thank you and it has been added as “However, the current approved vaccines have either been associated with safety concerns or are in limited supply.”

Line 24 Please add what virus (mpox? VACV?)

VACV has been added.

Line 37 Is this indicating that vaccination with live VACV ended in the 1970s? I believe some countries continued to vaccinate into the 1980s, please confirm. Additionally military (in some countries) and lab workers have continued to be vaccinated with VACV.

Thank you for your input. Although we did not find any published reports of smallpox vaccinations in the 1980s, we did uncover some evidence suggesting that certain countries continued to mandate vaccination certificates for international travelers post-1980 (see WH_1980_May_p27-29_eng.pdf on the WHO website).

Line 44 JYNNEOS is MVA (just with the tradename JYNNEOS). Is there another version of MVA (besides JYNNEOS) available commercially in non-US countries?

Thanks, it has been corrected. Our data search yielded information indicating that JYNNEOS is the only version of MVA, known internationally as Imvanex/Imvamune®.

Line 47 this is not correct for ACAM2000, which is one of the VACV vaccines. It is replication competent. Please correct this sentence

Thanks. It has been revised to read as follows: " There are certain safety concerns associated with the attenuated virus vaccine ACAM2000, particularly in individuals with immunodeficiency."

Line 48 the data thus far from the mpox outbreak suggests it is Very safe. This paper also has that conclusion: <https://pubmed.ncbi.nlm.nih.gov/30445121/> Suggest instead of safety, the focus is on mRNA vaccine production may be more quickly ramped up than the live vaccines.

Thanks for your suggestion. It has been revised as “Moreover, the limited availability of JYNNEOS vaccine underscores the pressing need for a vaccine that can be developed quickly. “

Line 59 Is this from previous studies? please add for clarity

Yes, the reference has been added. doi:10.1128/JVI.65.10.5631-5635.1991

Line 60 Mice are insusceptible to smallpox and this work can only be done at two locations in the world (not by this lab cited). I believe this is also a VACV challenge, please confirm and correct.

Yes, it was corrected to VACV.

Line 60 change to vaccinated

Thanks, it has been corrected.

Line 68 required for protection?

Yes, as that paper said, "while for lethal intranasal challenge, both L1R and A33R are required for protection."

Line 75 Is there any information on these proteins within MPXV since there are some differences in proteins between OPXVs?

Until so far, we did not find any structural or functional information of M1R and A35R.

Line 140 please clarify that this is VACV PRNTs, not MPXV

Yes, VACV was added.

Line 147 unclear in all of these studies why studies were terminated so early

As suggested, we repeated all the in vivo studies and extended the observation time to at least two weeks.

Line 206 In the figure legend below, there is no reference to mice vaccinated with VACV-WR at two sublethal doses (although the figures have a VACV high and low group, assuming that is these animals?). Instead the figure refers to animals receiving passive immunity ("For passive protection, three naïve mice in each group received 100 µL sera (IP) from the mice of 7-day post vaccination ") Where are these animals in the figures?

The VACV-low and -high groups were vaccinated with VACV-WR at two sublethal doses. Sorry for the confusion, and the passive protection legend had been moved to Figure 6.

In our new repeated experiments, we did these experiments separately, so it is clear now, please see Figure 4 and 6.

Line 246 Same comment as above. I think this figure legend needs to be corrected to remove reference to passive protection group and add the low/high vaccination with VACV WR, or otherwise edited as it is unclear right now

Thanks. It has been corrected as commented.

Line 246 Figure 6B?

In our revised manuscript, the passive protection data are shown in Figure 6.

Line 297 Provide references here. And since mice are insusceptible to VARV (unless they are humanized), are you referring to VACV studies? Please correct the sentence and provide references.

Sorry for the oversight. They did use VACV for in vivo studies and VARV for some in vitro neutralization study. It has been corrected.

Line 308 Limitations and future experiments should be included. The study focused on VACV (both the in vitro work and the in vivo studies). Studies with MPXV as the antigen in PRNTs as well as MPXV in vivo studies should be performed. Also the study did not compare vaccination using MVA/JYNNEOS (where the known "peak immune response following the second dose" could have been used in the study design, but instead a different VACV.

Thanks. Since we have no access to MVA/JYNNEOS at the present time, the limitations were

discussed in the end of discussion part.

Line 394 Add whether anesthesia was used and the volume of inoculum

Thanks. They were added.

Line 396 15% weight loss is not usually used for euthanasia. It is usually 20-25% (in some pubs even 30%). Some mice may have been euthanized that would have survived infection. Likewise, 9 days pi is rather early to end the study. It would have been a stronger study if it went out to the typical ~14-20 days for this model.

Our animal welfare committee typically uses a body weight loss threshold of 15%-20% as the point for euthanization. However, after discussing this study with the committee, we received authorization to allow for a transient body weight loss of up to 30%. To ensure reliable and accurate results, we extended the observation time to at least two weeks post-virus challenge in all repeated experiments.

REVIEWERS' COMMENTS

Reviewer #1 (Remarks to the Author):

The authors have addressed all my comments appropriately.